# Classification of mountain-based rail transit stations and analysis of passenger flow influencing mechanisms

Qingru Zou[1,2]*, Yue Xia[2], Xinchen Ran[1,2], Xueli Guo[2], Jiaxiao Feng[1,2]

1 Department of Chongqing Key Laboratory of Intelligent Integrated and Multidimensional Transportation System, Chongqing Jiaotong University, Chongqing, China, 2 Department of School of Traffic and Transportation, Chongqing Jiaotong University, Chongqing, China

* 990201900066@cqjtu.edu.cn

## Abstract

Mountainous urban rail transit stations exhibit distinct characteristics. To investigate how these features affect passenger flow variations at rail stations, we analyze geographic-environmental data surrounding the stations and integrate road network topology, automatic fare collection data, and point-of-interest (POI) data. We propose a method to classify rail transit stations by considering the mountainous features and establish a multiscale geographically weighted regression (MGWR) model to assess the classification results. This study focuses on 189 rail stations in Chongqing, identifying six station categories: comprehensive mountainous, comprehensive non-mountainous, employment mountainous, employment non-mountainous, residential mountainous, and residential non-mountainous. The MGWR results show that road growth coefficients, average longitudinal slopes, and road lengths significantly influence station performance. For instance, the average longitudinal slope substantially affects employment in mountainous stations, particularly during the morning peak. The analysis reveals that the average longitudinal slope exerts a stronger negative effect on morning peak inbound passenger flow at employment mountainous stations (-0.949), indicating that commuters are more sensitive to travel time during the morning peak. In contrast, the evening peak inbound passenger flow is less impacted (-0.409), suggesting that evening commuters face fewer time constraints. These findings offer strategic insights for zoning transit stations to support transit-oriented development(TOD).

## 1 Introduction

As economies develop, traffic congestion in major cities is becoming increasingly severe. To address this issue, priority has been given to developing urban public transportation systems, particularly by promoting urban rail transit as the city's public transportation backbone, supplemented by regular buses. For example, in Tokyo,

**Data availability statement:** All relevant data are within the manuscript and its Supporting information files.

**Funding:** This study is jointly supported by the National Natural Science of China [grant number: 52302386], and the China Postdoctoral Science Foundation [grant number: 2023M730430]. All funds were received by Qingru Zou. The funders had no role in study design, data collection and analysis, decision to publish, or preparation of the manuscript.

**Competing interests:** The authors have declared that no competing interests exist.

Japan, rail transit accounts for as much as 77.7% of the total public transportation usage [1], and in Paris, France, it accounts for 70% [2], and in Shanghai, China, it reaches 77.5% as well [3]. However, in mountainous cities, due to complex spatial geography and urban road layout constraints, traffic congestion is even more pronounced, and the role of rail transit as a backbone is less apparent. In Seoul, South Korea, rail transit accounts for 36.2% of total public transportation usage, and in Chongqing, China, it stands at 40.1% [4].

The main reasons why rail transit plays an insignificant role in the backbone of mountain cities are as follows: First, due to the steep gradients of mountainous roads and the limited capacity of rail systems, such as Chongqing's monorail Line 3, which has low capacity, fewer doors, and long stop times, the morning peak capacity is constrained [5]. Second, public-to-rail transfers around stations are inefficient. Mountainous cities tend to have multicentric clusters, and rail stations are often located on the periphery of districts, making passenger flow distribution less efficient and necessitating ground transport for the "last mile" [6].Lastly, the walking paths around mountainous rail stations are complex, with high road growth coefficients, resulting in poor pedestrian accessibility [7].

Therefore, to mitigate the influence of mountainous spatial characteristics on passenger flow and enhance the pivotal role of rail transit in mountainous cities, it is imperative to study the factors affecting passenger flow in mountain rail transit and increase the utilization rate of urban rail transit in such regions. At present, when analyzing the impact of passenger flow on rail transit stations [8], scholars rarely consider mountainous characteristics, even though spatial differences in influencing factors exist. This paper focuses on mountainous features such as the average longitudinal slope of roads, road network length, and road growth coefficient. It mainly examines walking accessibility around rail stations, transfer convenience, surrounding development intensity, and station passenger flow data. The K-means clustering method is applied to classify Chongqing's rail transit stations. Subsequently, OLS, GWR, and MGWR models are established to analyze the stations using different classifications.

## 1.1 Literature review

Since the characteristics of each station and its surrounding land use affect stations differently, leading to variations in passenger flow patterns, it is essential to investigate the mechanisms influencing passenger flow across different station types based on a reasonable classification of stations.

### 1.1.1 Classification of urban rail rapid transit stations.
Selecting appropriate station classification indicators is key to the classification of rail transit stations. Existing research can be divided into single-indicator and multi-indicator classification approaches. The single-indicator classification includes:

(1) Classification of station characteristic indicators: Wu Jiaorong (2007) categorized subway stations into distinct functional types based on their varying transfer facilities [9].

(2) Land use index classification: In the United States, the Center for Transit-Oriented Development (2010) classified stations into 15 types by considering factors such

as the number of jobs around the stations, the annual total mileage of households in the traffic area, and whether the surrounding areas were residential, employment-focused, or mixed-use [10].

(3) Passenger flow index classification: Meekyung (2018) grouped 233 subway stations in Seoul into eight categories using data on subway card swipes, passenger flow at different times, and fluctuations in passenger flow curves [11]. Similarly, Li et al. classified stations into six categories based on passenger traffic characteristics such as the number of peaks and troughs and the skewness in flow fluctuations [12].

In recent years, more researchers have combined multiple indicators to classify stations. For instance, Higgins. et al used data from 372 rail transit stations in Toronto, Canada, to extract land use and passenger flow characteristics, classifying them into ten categories [13]. Kim et al. combined passenger flow and land use data to classify Seoul's stations [14]. Similarly, Pang Lei et al. classified Tianjin's rail transit stations into three categories—residential, employment, and mixed-use—by combining built environment, station attributes, and network characteristics [15]. Chen L. et al.used station characteristics and nearby building functions to categorize 30 stations in Xi'an into six types [16].

**1.1.2 Study on the influence mechanism of passenger flow.** Regression analysis is a common method to determine how different indicators affect passenger flow at rail stations. Traditional models such as Ordinary Least Squares (OLS) have been widely used. For example, Loo et al. used the OLS model to analyze rail transit stations in Hong Kong and New York, concluding that various indicators within the service areas significantly impacted station passenger flow [17]. An et al. employed Ordinary Least Squares (OLS) regression analysis and found that commercial land use, bus stations, and tourist attractions have significant positive impacts on Shanghai's rail transit passenger flow, independent of weekdays [18]. However, with the deepening of research, some scholars argue that the OLS method lacks consideration of spatial spillover effects. The Geographically Weighted Regression (GWR) model addresses this by visualizing how independent variables affect dependent variables across different spatial locations. For instance, Qian et al. found that the GWR model offered better explanatory power when studying the dynamic demand for cabs in New York City [19].

The GWR model, however, assumes that all independent variables affect the dependent variable at the same spatial scale, ignoring spatial heterogeneity. To address this limitation, researchers introduced the MGWR model. Jun et al.applied the MGWR model to Seoul's rail stations, finding that it effectively accounted for both global and local variables [20]. Zahratu et al. highlighted that the MGWR model allows each independent variable to have a specific bandwidth, reflecting the differentiated scales of influence [21]. Tak et al. used the MGWR model to analyze passenger flow in Beijing's rail stations, concluding that it provided more reliable estimates compared to the classical OLS and GWR models [22].

In summary, while considerable progress has been made in classifying rail transit stations and understanding passenger flow mechanisms, further improvements are necessary. First, much of the existing research focuses on plain areas, overlooking the unique geographic conditions of mountainous cities. Additionally, many studies fail to consider differences in the factors affecting passenger flow across various types of stations. Therefore, it is critical to develop a reasonable classification system for rail stations and examine the passenger flow mechanisms of different station types, considering the spatial and geographic characteristics of mountainous cities, to enhance the rail transit share in such areas.

## 1.2 Study area

Chongqing has a complex geographic environment with significant elevation changes, making it a typical mountainous city in China. Except for certain sections of monorail Line 3, which reaches full capacity during peak hours, the other lines maintain ample capacity [23]. This paper selects nine operational rail transit lines in Chongqing, comprising Lines 1, 4, 5, 6, 10, the Circular Line, and the Guobo Line as subway systems (high-capacity rail transit), and Lines 2 and 3, which employ the straddle-type monorail technology (medium-capacity light rail transit system). The study investigates a total of 189 stations across these lines, excluding repeated interchange stations. It integrates Chongqing's road network structure

and pedestrian system characteristics, constructing a "key impact area" centered around each rail transit station, with 500-meter radius, known as the station area [24].

## 2 Data and methodology

### 2.1 Cluster-based identification of station types

Analyzing passenger flow characteristics at urban rail transit stations is a relatively complex endeavor [25]. Classifying different stations can simplify the research.The clustering methods currently employed by scholars include K-means, DBSCAN, HDBSCAN, OPTICS, and Self-Organizing Maps (SOM). DBSCAN performs poorly on high-dimensional data and is ineffective for clusters with varying densities. When the data set exhibits significant spatial density variations, it struggles to reasonably cluster both high-density and low-density regions. HDBSCAN is highly sensitive to parameter settings and has a high computational complexity. The OPTICS algorithm also suffers from high computational complexity, especially on large-scale data sets, and faces difficulties in extracting meaningful clusters. Self-Organizing Maps (SOM) require the selection of an appropriate network structure, and improper choices can negatively impact model performance. Although the K-means algorithm is sensitive to initial cluster centers and faces challenges in determining the optimal number of clusters (K), its simplicity, efficiency, and suitability for large-scale data sets make it a rational choice for our study. We determine the optimal number of clusters using the elbow method and improve the stability of clustering results by enhancing the initialization method. Moreover, K-means has fewer parameters, which are easy to adjust and interpret, an important advantage in practical applications. We believe that, with appropriate data preprocessing and parameter tuning, K-means can effectively classify mountain railway stations and provide support for the analysis of passenger flow impact mechanisms. The unsupervised classification of station types using the K-means clustering algorithm involves three key steps: selection of indicators, determining the number of clusters, clustering the stations, and identifying station categories.

**2.1.1 Selection of station classification indicators.** Based on station functionality, service capacity, and environmental impacts of surrounding areas, and considering data accessibility, quantifiability, and comprehensiveness, we propose four categorized indicators as classification criteria.Development intensity indicator--reflecting urbanization levels and economic activity density in station-adjacent areas;Transfer convenience--indicating accessibility to other zones and the station's functional role within the transportation network;Walking accessibility--measuring passenger convenience in reaching stations;Passenger flow--capturing distinct temporal characteristics of ridership during morning/evening peak periods.

(1) Development intensity indicator. POI data contain spatial information of the urban fabric, including attributes such as names, category classifications, and geographical coordinates (latitude and longitude). These data can effectively reveal the density and distribution of developments around these stations. The types of facilities and service functions within a 500-meter radius around the stations significantly influence the attractiveness and vitality of the transit stations [26]. This study employs Gaode Map to obtain POI data within a 500-meter radius around rail transit stations in nine districts of Chongqing, namely Yuzhong, Shapingba, Jiangbei, Yubei, Nan'an, Jiulongpo, Dadukou, Banan, and Beibei. Based on the "Urban Land Use Classification and Planning and Construction Land Use Standards," a total of 298,035 POIs were extracted from the core station area [27]. This includes 138,887 POIs for commercial land use, 72,878 for residential land use, 40,215 for public administration and services, and 23,766 for transportation purposes.

(2) Transfer convenience indicator. The number of bus lines within 500 meters of the station is calculated based on Gaode POI data to measure the ease of transferring between buses and rail transit [28]. Additionally, the number of parking lots is used to reflect the convenience of Park+Ride (P+R) modes. A total of 3,585 bus lines and 3,316 parking lots around the stations are extracted.

(3) Walking accessibility indicators [29]. The 500-meter radius around rail transit stations is generally regarded as the core area for pedestrian accessibility. Within this range, residents and passengers can conveniently reach the stations on foot without relying on other transportation modes. The characteristics of roads and buildings within this range directly influence the walking experience and travel efficiency of passengers, making it a focal area for research. Using Open Street Map data, walking accessibility is measured by the length of the road network within a 500-meter radius around the station area. The degree of zigzagging and elevation differences in the terrain are reflected by the road growth coefficient and average longitudinal slope, as shown in equations (1 and 2). A total of 3,842 road segments were extracted, allowing the calculation of road length, growth coefficient, and average slope.

$$I = \frac{H - H_o}{L_t} \times 100\%$$

(1)

Where:

I is the average longitudinal slope, H is the station elevation, $H_o$ is the passenger departure point's elevation within 500m, and $L_t$ is the length of the passenger trip within 500m.

$$C_r = \frac{L_r}{L_s}$$

(2)

Where:

$C_r$ is the road growth factor, $L_r$ is the length of the passenger walk within 500m, and $L_s$ is the straight-line distance from the start to the end of the passenger walk.

(4) Station Passenger Flow Indicator: The station passenger flow data from the urban rail transit automatic fare collection system in 2021. The morning and evening peak inbound and outbound passenger flow of each station is selected to reflect the passenger flow changes of passengers in different periods [30]. The station passenger flow indicators are derived from the AFC data in 2021.

Thirteen indicators were selected to classify stations based on four categories of factors. To reduce inconsistencies in scale between different indicators, data were standardized using the Z-score method. The Z-score method standardizes data by subtracting the mean of each variable and dividing by its standard deviation. This process ensures that all variables have a mean of 0 and a standard deviation of 1, thus eliminating the influence of scale differences (as shown in Table 1).

**2.1.2 Determination of the number of clusters.** Before conducting cluster analysis, it is necessary to determine the appropriate number of clusters. This study applies the "elbow method" to determine the optimal number of clusters by analyzing the sum of squares of errors (SSE) for various cluster sizes and identifying the elbow point where the marginal improvement in SSE diminishes (i.e., the inflection point). The optimal number of clusters is identified by observing the largest reduction in SSE, corresponding to the inflection point. The basic principle behind the "elbow rule" is that as the number of clusters (k) increases, the clustering becomes more granular, and the cohesion within each cluster improves, causing the SSE to decrease. However, after a certain point, further increases in k result in diminishing improvements in cohesion, causing the SSE reduction to slow. This produces an "elbow" shape on a graph plotting SSE against k, and the k-value corresponding to this elbow is considered the optimal number of clusters. The SSE is calculated using the following formula:

$$SSE = \sum_{i=1}^{k} \sum_{p \in C_i} |p - m_i|^2$$

(3)

**Table 1. Indicators for station classification.**

| Type of indicator | Specific disaggregated indicators | Data range before standardization | Range of standardized data |
|---|---|---|---|
| Development intensity | Business POI | [22, 6395] | [-0.7391, 5.4198] |
| | Residential POI | [8,4448] | [-0.6052,5.9111] |
| | Public Administration POI | [0,1884] | [-0.7337,5.8002] |
| | Transportation POI | [0,1572] | [-0.6867,5.7129] |
| Transfer convenience | Public transport line | [2,58] | [-1.5202,3.8815] |
| | Parking lots | [1, 88] | [-0.8182, 4.9127] |
| Walking accessibility | Length of road network | [53.794,543.755] | [-0.8826,5.6359] |
| | Road growth factor | [-101.88, 35.54] | [-1.7907, 2.6959] |
| | The average longitudinal slope of the road | [1, 41.97] | [-5.3308, 1.9317] |
| Station passenger flow | Morning peak inbound passenger flow | [11,13374] | [-1.1327,4.4847] |
| | Evening peak inbound passenger flow | [15,21243] | [-0.8485,5.6697] |
| | Morning peak outbound passenger flow | [9,16914] | [-0.7855,6.6339] |
| | Evening peak outbound passenger flow | [9,15875] | [-0.9432,6.7220] |

where:

K is the current number of clusters; $C_i$ is the ith subset in the clustering; p is the sample points within the subset; and $m_i$ is the average of all sample points within the subset.

## 2.2 Influence mechanism of passenger flow in different station types

### 2.2.1 Selection of indicators for passenger flow impact mechanisms.

(1) Station functional attributes

The operational and geographical attributes of urban rail transit stations influence passenger flow [31]. Based on passenger flow statistics, this study designates stations with particularly high passenger volumes at transfer points as large transfer stations. The geographical location of the station is used to determine whether it serves as an external transportation hub and whether it is in proximity to a large commercial area. These three indicators are collectively referred to as the station functional attributes indicators of the station.

(2) Built environment

In addition to the previously mentioned indicators, the land use mix index was also considered, reflecting the diversity of land use types in the station area [32].

$$H_{(X)} = -\sum_{i=1}^{n} P_i \log P_i$$

(4)

Where:

$H_{(X)}$ is the land use mix index at station X; $P_i$ is the percentage of the number of types i POIs at station X to the total number of POIs at that station.

This paper analyzes two dimensions: the built environment and the station's attributes. Among them, three indicators, whether the station is an interchange station, whether it is an external transportation hub, and whether it is adjacent to a large business district, are discrete, while the rest are continuous variables. Table 2 presents an overview of these potential influences. The characteristics of the station itself were obtained through observation and the compilation of

**Table 2. Factors affecting passenger flow.**

| Form | Variable name | Abbreviation | Description |
|---|---|---|---|
| Staion functional attributes | Transfer station | TRANS | 1 yes, 0 no |
| | External Transportation Hub | HUB | 1 yes, 0 no |
| | Neighborhood | STO | 1 yes, 0 no |
| Built environment | Length of road network | RO-LE | Total length of passenger walking paths within the station's catchment area |
| | The average longitudinal slope of the road | RO-APG | The average gradient of passenger walking paths within the station coverage area |
| | Road growth factor | RO-GO | The ratio of straight-line distance to the station for passenger trips within the station's coverage area to the actual distance to the station |
| | Number of parking lots | PARKING | Number of parking lots within the station |
| | Number of bus routes | BUS-LINE | Number of bus routes within the station coverage area |
| | Number of commercial POIs | POI-STO | Number of commercial land POIs within the station coverage area |
| | Land use mixing ratio | LAND-MIX | Mixed rate of land use within the station |
| | Number of residential POIs | POI-RES | Number of residential land POIs within the station coverage area |
| | Number of public administration POIs | POI-AMD | Number of POIs for public administration and public services within the station coverage area |
| | Number of transportation POIs | POI-TRANS | Number of transportation POIs within station coverage |

web resources, while the built environment was based on categorical data with the addition of a land use mixing rate, as shown in Table 2.

**2.2.2 Selection of key variables.** Only if there is no strict collinearity between the independent variables can the model analysis be conducted. The collinearity test is represented by the variance inflation factor (VIF). The greater the VIF, the higher the collinearity probability between the independent variable and other variables.VIF within 10 indicates that there is no serious collinearity [33].

The VIF of the jth independent variable is:

$$VIF_j = \frac{1}{1-R_j^2}$$

(5)

Where:

$VIF_j$ is the variance inflation factor of the jth independent variable; $R_j^2$ is the determination coefficient of the jth independent variable as the dependent variable, and the other independent variables are used as linear regression.

There may be spatial dependence between neighboring stations, and this influence cannot be explained by OLS regression, so it is necessary to test the spatial correlation between stations. In this study, Moran's I was selected as the test index, and the spatial projection coordinates and influencing factors of each station were imported into Arc GIS for the spatial autocorrelation test [34]. Prior to the calculation of the Moran's I index, the concept of a spatial weights matrix must be introduced. This matrix reflects the spatial proximity between multiple locations by representing the relationships among different spatial units. In the context of the Moran's I index, the spatial weights matrix is often constructed using Euclidean distance as the basis for assigning weights. The product of the weight matrix γ and the explanatory variable indicators, as shown in Equation (10), reflects the degree of similarity between spatial units. Euclidean distance, which measures the straight-line distance between two points, provides an accurate representation of the actual distance between them in geographical space and thus serves as a valid basis for assigning weights.

The calculation formula of Moran'I is as follows:

$$\text{Moran'I} = \frac{\sum_{i=1}^{m'} \sum_{j=1}^{m'} \gamma_{ij}(e_i - \bar{e})(e_j - \bar{e})}{S^2 \sum_{i=1}^{m'} \sum_{j=1}^{m'} \gamma_{ij}} \tag{6}$$

$$\gamma_{ij} = e^{-\left(\frac{d_{ij}}{b}\right)^2} \tag{7}$$

Where:

m' is the total number of rail transit stations; $\gamma_{ij}$ is the weight between stations; $e_i$ and $e_j$ are represented as independent variable indicators for the i and j stations; $\bar{e}$ is the mean value of station- independent variable indicator e; $S^2$ is the variance of the station-independent variable index e; $d_{ij}$ is the Euclidean distance between stations i and station j; b is the bandwidth, which refers to the non-negative decay parameter of a weighted remote function.

When the sample space distribution is relatively uniform, equal bandwidth is often used, and the opposite is used.The value of Moran'I is usually [-1, 1]. Under certain significance tests, the Moran index is 0. That is, there is no spatial auto-correlation. A value greater than 0 indicates that the variable has agglomeration, while a value less than 0 indicates that the variable has dispersion (spatial negative correlation) [35]. Generally, normalized statistical Z-value is used to conduct a significance test, and its expression is as follows:

$$Z(\text{Morans'I}) = \frac{\text{Morans'I} - E(\text{Morans'I})}{\sqrt{\text{Var(Morans'I)}}} \tag{8}$$

Where:

E (Moran'I) is the theoretical mathematical expectation of the Moran'I; $\sqrt{\text{Var(Morans'I)}}$ is the theoretical variance.

The reliability of Moran'I can be evaluated by the Z-value and P-value of significance. If the significance P value is less than 0.05 (through a 95% confidence test) the absolute value of the Z score exceeds the critical value of 1.96, indicating that the results of Moran'I are credible, and more than 95% of the certainty is that the data are spatially correlated. When the value of Moran'I is greater than 0, the spatial correlation is positive; when the value of Moran'I is less than 0, the correlation is negative [36].

**2.2.3 Passenger flow impact regression modeling.** The OLS, GWR, and MGWR models are commonly used to handle continuous dependent variables. When analyzing the factors influencing station passenger flow in regression models, passenger flow data can be treated as a continuous variable.In the regression models, the dependent variables include the total daily passenger flow, morning peak inbound passenger flow, morning peak outbound passenger flow, evening peak-hour inbound passenger flow, and evening peak outbound passenger flow. The independent variables are the station functional attributes and built environment indicators introduced in the previous section.

(1) OLS Regression Model [37]

$$Y = \alpha_0 + \alpha_1 x_1 + \alpha_2 x_2 + \cdots \alpha_n x_n + \varepsilon \tag{9}$$

Where:

$X_1$, $X_2$,..., $X_n$ are the independent variables; $\alpha_0$ is the intercept term of the OLS model, $\alpha_1$, $\alpha_2$,..., $\alpha_n$ are the regression coefficients for the nth independent variable; Y is the dependent variable; and ε is the residual, which follows a normal distribution with a mean of zero.

(2) GWR regression model [38]

$$Y_i = \beta_{0(U_i,V_i)} + \sum_{j=1}^{n} \beta_{j(U_i,V_i)} X_j + \varepsilon_i \tag{10}$$

Where:

$(U_i, V_i)$ is the coordinates of station i; $Y_i$ is the dependent variable at location i; $\beta_{0\,(U_i,\,V_i)}$ is the intercept of station i; $\beta_{j\,(U_i,V_i)}$ is the regression coefficient of the jth independent variable at $(U_i,V_i)$; and $\varepsilon_i$ is the residuals of station model i.

(3) MGWR regression model [39]

$$Y_i = \beta_{0(U_i,V_i)} + \sum_{l=k+1}^{n'} \beta_{l(U_i,V_i)} e_l + \sum_{l=1}^{k} \beta_l e_l + \varepsilon_i \tag{11}$$

Where:

$\beta_{0\,(U_i,V_i)}$ is the intercept for the station i; $\varepsilon_l$ is the lth global indicator for the station; $\beta_l$ is the regression coefficient for the global indicators, which are the first k indicators; and $\beta_{l\,(U_i,V_i)}$ is the regression coefficient for the local indicators, i.e., the k+1 to n' indicators, which vary with each station.

## 3 Analysis of results

### 3.1 Station classification results

K-means clustering is an iterative analysis method that achieves optimal classification by continuously adjusting the positions of cluster centers [40]. The station data is divided into K groups, randomly select K objects as the initial cluster center, and then calculate the distance between each object and each seed cluster center, and assign each object to the nearest cluster center, the cluster center, and the object assigned to them represent a cluster. For each sample assigned, the cluster center of the cluster is recalculated based on the existing objects in the cluster. This process is repeated until no objects are reassigned to different clusters.

The number of clusters was determined using the elbow method. Python code was used to analyze the data samples and identify the clustering type (K) [41], as shown in Fig 1. The analysis reveals that as the K value increases from 1 to 6, there are clear fluctuations in the degree of aberration. When the K value exceeds 6, the variation in aberration decreases significantly. Thus, 6 was selected as the optimal number of clusters.

For the six categories of stations in the clustering results, Figs 2 and 3 show the inbound and outbound passenger flows for each type of weekday station and the characteristics of each type of station.

(1) Comprehensive-type stations exhibit the highest passenger volumes, with weekday traffic showing distinct bimodal patterns. Comprehensive mountainous stations have higher longitudinal slopes and road growth coefficients. In contrast, Comprehensive non-mountainous stations have higher passenger flows and lower slopes, with more convenient land use and transport conditions.

(2) Employment-type stations feature higher passenger flows, with weekday flow curves showing a single-peak pattern, with significantly higher outbound flows in the morning peak hour and inbound flows in the evening peak hour. Employment mountainous stations have a higher percentage of POI for commercial facilities and services and a larger number of bus stops and parking lots in the vicinity. Employment non-mountainous stations have higher passenger flows than employment mountainous stations, have gentler topography that is more conducive to urban planning and land development and utilization around the station, have high road network density, and have a higher density of bus routes within the station coverage area.

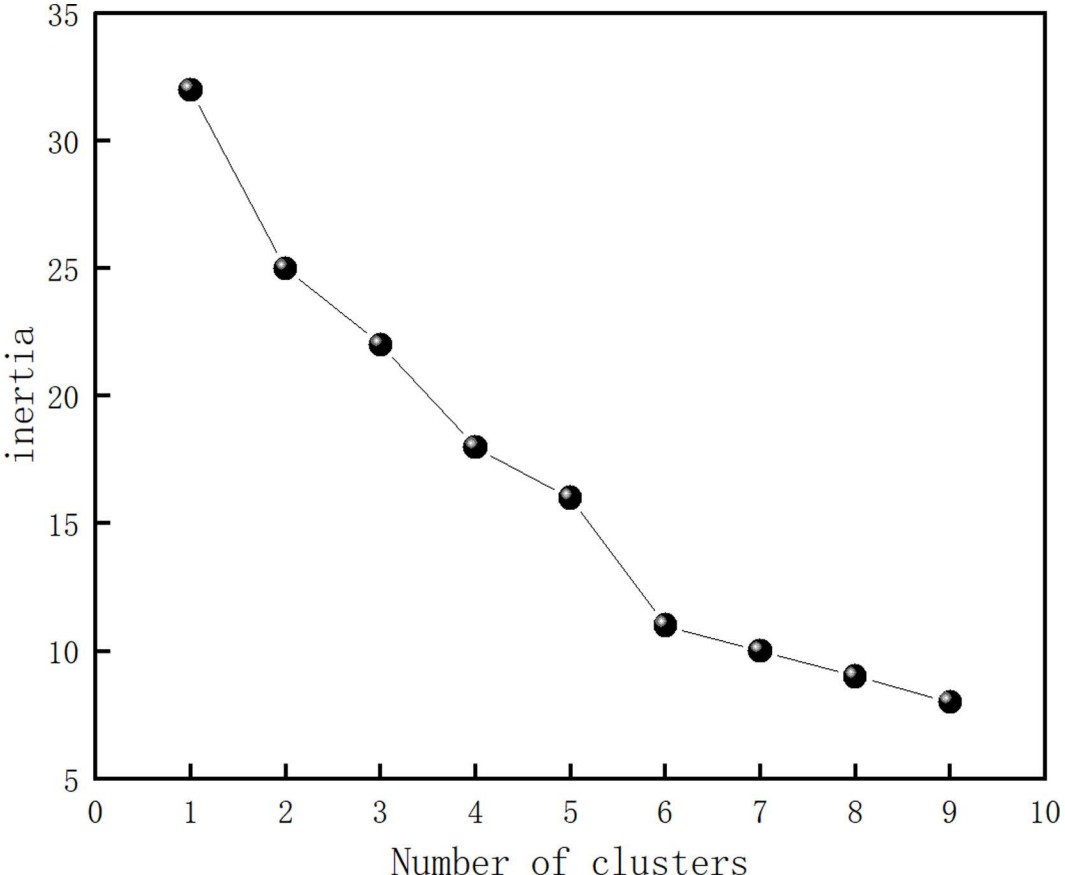

**Fig 1. Error sum of squares for different numbers of clusters.**

(3) The residential-type stations exhibit the smallest passenger flow, with a single-peak pattern of weekday passenger flow, and large inbound passenger flow in the morning peak and outbound passenger flow in the evening peak. The geographical composition of the surrounding area of the residential mountain-type stations is dominated by residential land, and the degree of development of the surrounding land is not high. The passenger flow at residential non-mountainous stations is larger than that at residential mountainous stations, the road network density is high, the average longitudinal gradient is low, and the POI of residential stations is high. This indicates the characteristics of residential riders who leave early and return late.

After classifying 189 stations in Chongqing, by analyzing the passenger flow entering and leaving different types of stations and the characteristics of stations around the stations, it was found that there were significant differences among different types of stations. To further explore the relationship of the influence of different influencing factors of the stations on the passenger flow of the stations, a passenger flow regression model can be established to analyze the influence of the influencing factors on the stations. Propose improvement measures for different types of sites to increase passenger flow.

### 3.2 Passenger flow impact regression model results

**3.2.1 Results of key variables selection.** The covariance test results and spatial autocorrelation of the factors influencing the passenger flow characteristics of Chongqing rail transit stations are shown in the Table 3.

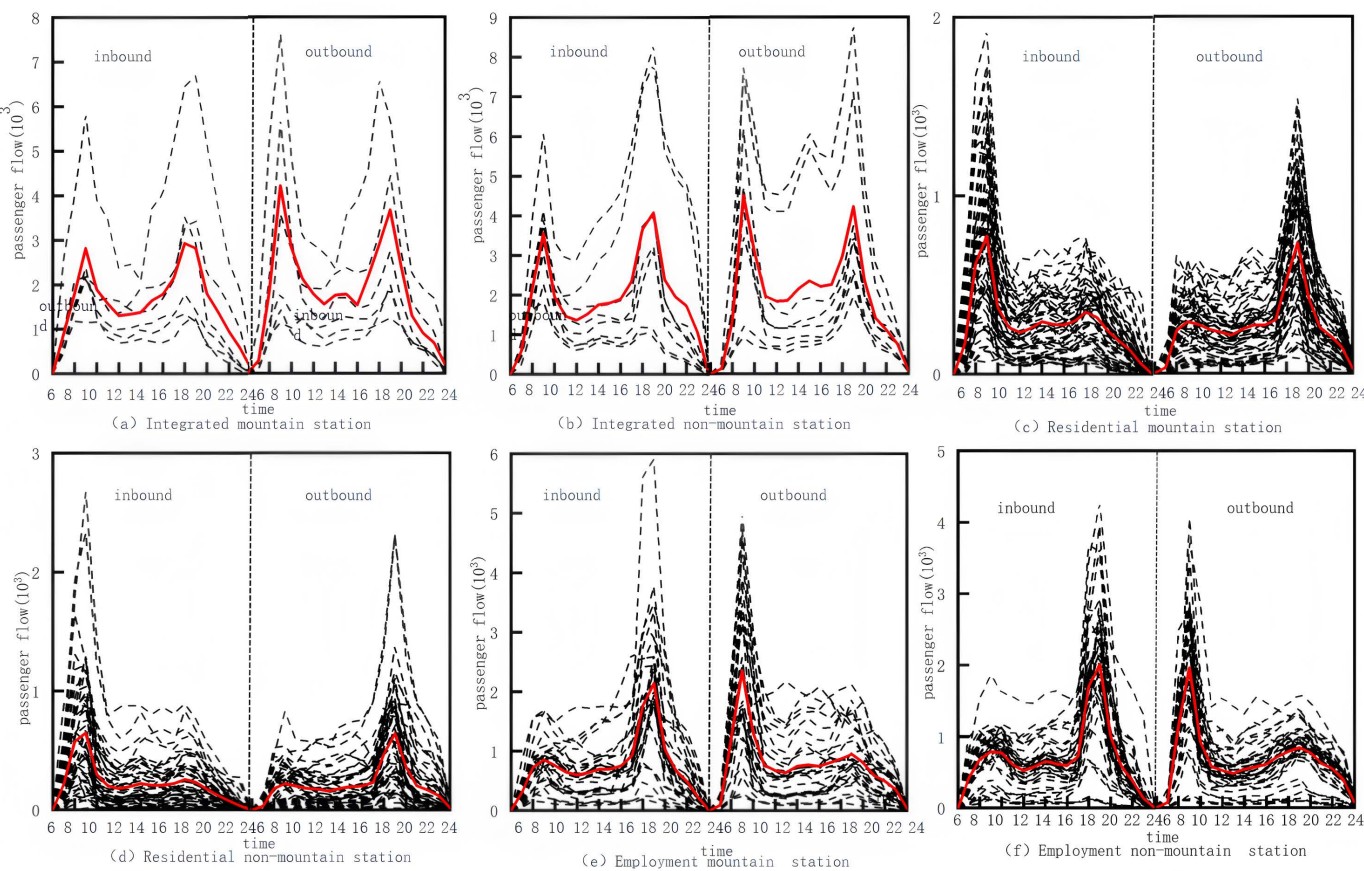

**Fig 2. Passenger flow in and out of different types of stations.**

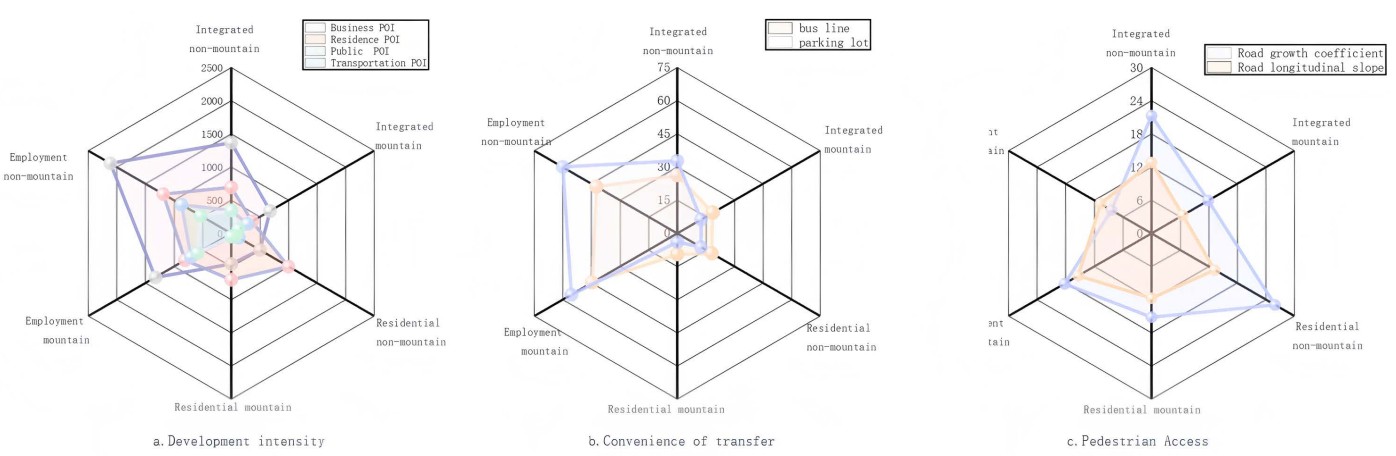

**Fig 3. Characteristics of different types of stations.**

**Table 3. Factors affecting passenger flow.**

|  | Variable indicators | Average value | (Statistics) standard deviation | VIF | Moran'I | Z-value |
|---|---|---|---|---|---|---|
| Station functional attributes | TRANS | N/A | N/A | 1.452 | 0.187 | 8.231 |
|  | HUB | N/A | N/A | 1.392 | 0.163 | 7.11 |
|  | STO | N/A | N/A | 1.603 | 0.416 | 17.583 |
| Built environment indicators | RO-LE | 120.135 | 75.164 | 1.779 | 0.233 | 10.214 |
|  | RO-APG | -1.011 | 18.921 | 1.145 | 0.116 | 5.287 |
|  | RO-GO | 16.973 | 9.271 | 2.525 | 0.311 | 13.593 |
|  | PARKING | 15.197 | 17.466 | 4.125 | 0.224 | 9.822 |
|  | BUS-LINE | 18.575 | 12.218 | 1.945 | 0.333 | 14.215 |
|  | POI-STO | 769.145 | 1038.017 | 8.311 | 0.313 | 13.522 |
|  | LAND-MIX | 0.398 | 0.196 | 1.217 | 0.075 | 3.353 |
|  | POI-RES | 413.109 | 682.600 | 2.714 | 0.021 | 1.153 |
|  | POI-AMD | 211.564 | 288.341 | 3.01 | 0.179 | 7.777 |
|  | POI-TRANS | 139.376 | 200.3556 | 3.889 | 0.165 | 7.124 |

The data show that the VIF values of the 13 indicators studied are all below 10, which means that there is no serious multicollinearity among these indicators and they have good validity. The Moran'I of the 13 indicators of the influencing factors are all positive. The Z-values of the 12 indicators are above 1.96, achieving the level of significance, which meets the conditions for establishing a regression model.

**3.2.2 Comparison of regression models.** The independent variables are regarded as global indicators in the OLS, while they are regarded as local indicators in the GWR, and the MGWR is a comprehensive model that integrates the global and local characteristics of the two models of OLS and GWR [42,43]. In this paper, we intend to take Chongqing metro stations as the research object, choose the same independent explanatory variables, and establish three methods of OLS, GWR, and MGWR to regress and analyze the passenger flow characteristics respectively.

The smaller the value of RSS, AIC, and AICc indicators, the larger the value of $R^2$ and adjusted $R^2$, the better the model fitting effect is [44]. As can be obtained from Table 4, the results of the three types of regression models show that:

**Table 4. Comparative analysis of passenger flow regression models.**

|  |  | RSS | AIC | AICc | $R^2$ | Adjusted $R^2$ |
|---|---|---|---|---|---|---|
| All-day weekday traffic | OLS | 74.057 | 389.074 | 393.817 | 0.612 | 0.584 |
|  | GWR | 65.006 | 384.359 | 391.646 | 0.66 | 0.613 |
|  | MGWR | 49.225 | 352.327 | 368.213 | 0.742 | 0.687 |
| Morning peak inbound traffic | OLS | 118.276 | 478.497 | 483.24 | 0.381 | 0.335 |
|  | GWR | 111.488 | 480.797 | 486.153 | 0.416 | 0.348 |
|  | MGWR | 97.388 | 470.691 | 481.225 | 0.49 | 0.403 |
| Morning peak outbound passenger flow | OLS | 105.164 | 456.054 | 460.797 | 0.449 | 0.409 |
|  | GWR | 97.142 | 454.489 | 459.846 | 0.491 | 0.432 |
|  | MGWR | 86.946 | 444.138 | 452.846 | 0.545 | 0.475 |
| Evening peak inbound traffic | OLS | 126.934 | 491.991 | 496.734 | 0.335 | 0.287 |
|  | GWR | 101.839 | 480.287 | 491.242 | 0.467 | 0.374 |
|  | MGWR | 72.812 | 437.826 | 459.687 | 0.619 | 0.52 |
| Evening peak outbound traffic | OLS | 80.148 | 404.171 | 408.914 | 0.58 | 0.55 |
|  | GWR | 73.524 | 402.148 | 407.739 | 0.615 | 0.569 |
|  | MGWR | 47.25 | 359.653 | 384.331 | 0.753 | 0.684 |

RSS, AIC, and AICc index values MGWR<GWR<OLS, $R^2$ and corrected $R^2$ values are MGWR> GWR>OLS, indicating that the MGWR and GWR models considering spatial correlation have a better fitting effect. Taking the AICc index as the bandwidth optimization criterion, it is found that the AICc index value of the MGWR model regression is significantly smaller than that of the GWR model in different periods, which indicates that the MGWR model realizes further optimization based on the GWR model.

**3.2.3 Influence mechanisms of passenger flow characteristics at stations based on MGWR.** To investigate the heterogeneous effects of mountain track characteristics, interchange convenience, and pedestrian accessibility on the passenger flow of different types of stations, based on the results of station classification, the MGWR model is used to study the influence mechanisms of all-day weekday passenger flow, morning peak inbound passenger flow, morning peak inbound and outbound passenger flow, evening peak inbound passenger flow and evening peak outbound passenger flow of different types of stations, respectively [45]. Take the employment class mountain type as an example, as shown in Table 5:

Table 5 shows that: (1) the coefficients of influence on the positive effect on the all-day passenger flow in the employment category of mountain-type stations are, in descending order, neighboring business districts (0.857)> commercial POIs (0.685)> parking lots (0.362)> public administration POIs (0.269)> traffic and transportation POIs (0.24)> external transportation hubs (0.121). It indicates that the commercial land use of this type of station and the neighboring business districts attract a large number of passengers and employment; (2) the coefficients of influence on the negative effect of all-day passenger flow are, in descending order, road growth coefficients (-0.613)>residential POIs (-0.429)>transit routes (-0.33)>mixed rate of land use (-0.281)>average longitudinal slopes of roadways (-0.181)>roadway network length (-0.1)> interchange (-0.033). The road growth coefficient has the largest negative impact, indicating that passengers at these stations are very sensitive to the length of the walk near rail stations, and has the second largest impact on residential POI, with the opposite travel behavior characteristics of the residential and employment class of stations having a large negative impact on employment hill traffic.

# 4 Conclusion

Based on passenger flow data and built environment data from mountainous urban rail transit stations, this study employs the K-means clustering algorithm to classify the stations and establishes OLS, GWR, and MGWR models for regression

**Table 5. MGWR model regression results of passenger flow characteristics for employment type hill type stations.**

| | Whole day | Morning peak inbound | Evening peak inbound | Morning peak outbound | Evening peak outbound |
|---|---|---|---|---|---|
| Length of road network | -0.1** | 0.416** | -0.329*** | 0.06** | 0.097** |
| The average longitudinal slope of the road | -0.181** | -0.949** | -0.409** | -0.087** | -0.055** |
| Road growth factor | -0.613** | -0.196** | 0.684** | -0.424** | -0.773** |
| Business POI | 0.685** | 0.821** | -0.049** | -0.377** | -0.995** |
| Residential POI | -0.429** | 0.425** | 0.705** | -0.333** | -0.903** |
| Public Administration POI | 0.269** | -0.731** | -0.84** | -0.188** | 1.067** |
| Transportation POI | 0.24** | 0.34** | 0.24** | 0.186** | 0.356** |
| public transport line | -0.33** | -0.134** | 0.009** | -0.103** | -0.365** |
| parking lots | 0.362** | -0.59** | -0.474** | 0.039*** | 0.675** |
| transfer station | -0.033** | 0.239** | 0.589** | -0.292** | -0.065** |
| external transportation hub | 0.121 | 0.004** | -0.142** | -0.154** | 0.007** |
| Neighborhood | 0.857** | 0.731** | -0.12** | 0.42** | 0.776** |
| Land use mixing ratio | -0.281** | 0.322** | 0.526** | 0.084** | -0.692** |
| $R^2$ | 0.763** | 0.632** | 0.86** | 0.945** | 0.808** |

Note: **"$p < 0.05$ (95% confidence level)".

analysis of passenger flow characteristics. The findings indicate that stations can be categorized into six types: comprehensive non-mountainous, comprehensive mountain, residential non-mountain, residential mountain, employment mountain, and employment non-mountain. A comparison of model performance reveals that the MGWR model achieves better regression fitting than the GWR and OLS models. The MGWR analysis highlights that the influence of various indicators on passenger flow is spatially non-stationary and varies with geographical location. For instance, a higher density of bus lines and parking lots around a station significantly increases peak passenger flow during the morning and evening. This effect is most pronounced at employment-oriented non-mountain stations and residential non-mountain stations. To enhance passenger flow, it is recommended to increase the number of buses and parking facilities near these types of stations. Additionally, lower average road slopes and road growth coefficients are associated with higher peak passenger flow, particularly at residential non-mountain stations. Thus, improving the pedestrian environment around these stations could attract more passengers.

This study provides a theoretical foundation for increasing the modal share of rail transit, though certain limitations remain. The passenger flow data used in this study were collected at hourly intervals. To better understand the influence of various indicators on passenger flow, future research could refine the time intervals to 15 minutes or less. Furthermore, while this study classifies POI data based on standard urban planning land use categories, subsequent research could further differentiate POI types to analyze the temporal and spatial distribution of passenger flow at urban rail transit stations with greater precision.

## Supporting information

**S1 File. All-day passenger flow.**
(XLSX)

**S2 File. Influencing factors.**
(XLSX)

**S3 File. Clustering algorithm.**
(PY)

**S4 File. Clustering algorithm.**
(PY)

**S5 File. Original POI.**
(XLSX)

**S6 File. Station functional attributes.**
(XLSX)

**S7 File. Morning and evening passenger flow.**
(XLSX)

**S8 File. Station latitude and longitude.**
(XLS)

## Author contributions

**Conceptualization:** Qingru Zou, Xinchen Ran.

**Data curation:** Qingru Zou.

**Formal analysis:** Qingru Zou.

**Funding acquisition:** Qingru Zou, Xinchen Ran.

**Investigation:** Yue Xia.

**Methodology:** Yue Xia.

**Project administration:** Qingru Zou.

**Resources:** Xinchen Ran.

**Software:** Xinchen Ran.

**Supervision:** Xinchen Ran, Jiaxiao Feng.

**Validation:** Xueli Guo.

**Visualization:** Xueli Guo.

**Writing – original draft:** Yue Xia.

**Writing – review & editing:** Yue Xia, Jiaxiao Feng.

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
