## [Decision Letter · Decision Letter 0]

11 Dec 2024

PONE-D-24-48526Classification of mountain-based rail transit stations and analysis of passenger flow influencing mechanismsPLOS ONE

Dear Dr. Zou,

Thank you for submitting your manuscript to PLOS ONE. After careful consideration, we feel that it has merit but does not fully meet PLOS ONE’s publication criteria as it currently stands. Therefore, we invite you to submit a revised version of the manuscript that addresses the points raised during the review process.

We look forward to receiving your revised manuscript.

Kind regards,

Qing-Chang Lu

Academic Editor

PLOS ONE

Journal Requirements:

When submitting your revision, we need you to address these additional requirements. 1. Please ensure that your manuscript meets PLOS ONE's style requirements, including those for file naming. The PLOS ONE style templates can be found at https://journals.plos.org/plosone/s/file?id=wjVg/PLOSOne_formatting_sample_main_body.pdf and https://journals.plos.org/plosone/s/file?id=ba62/PLOSOne_formatting_sample_title_authors_affiliations.pdf 2. Please match your authorship list in your manuscript file and in the system. 3. Please include a caption for figure 1 and 3. 4. Please note that PLOS ONE has specific guidelines on code sharing for submissions in which author-generated code underpins the findings in the manuscript. In these cases, we expect all author-generated code to be made available without restrictions upon publication of the work. Please review our guidelines at https://journals.plos.org/plosone/s/materials-and-software-sharing#loc-sharing-code and ensure that your code is shared in a way that follows best practice and facilitates reproducibility and reuse. 5. Thank you for stating the following financial disclosure:  [Funding: This study is jointly supported by the National Natural Science of China [grant number: 52302386],  and the China Postdoctoral Science Foundation [grant number: 2023M730430]. All funds were received by Qingru Zou. The funders had role in study design, data collection and analysis.].  Please state what role the funders took in the study.  If the funders had no role, please state: ""The funders had no role in study design, data collection and analysis, decision to publish, or preparation of the manuscript."" If this statement is not correct you must amend it as needed. Please include this amended Role of Funder statement in your cover letter; we will change the online submission form on your behalf. 6. Please upload a copy of Figure 1, 2, 3, and 4, to which you refer in your text on page 13 and 14. If the figure is no longer to be included as part of the submission please remove all reference to it within the text.

Additional Editor Comments:

There are important issues to be addressed as pointed out by the reviewers. Please provide detailed responses.

Reviewers' comments:

Reviewer's Responses to Questions

**Comments to the Author**

1. Is the manuscript technically sound, and do the data support the conclusions?

Reviewer #1: Yes

Reviewer #2: Yes

2. Has the statistical analysis been performed appropriately and rigorously? 

Reviewer #1: Yes

Reviewer #2: Yes

3. Have the authors made all data underlying the findings in their manuscript fully available?

Reviewer #1: No

Reviewer #2: Yes

4. Is the manuscript presented in an intelligible fashion and written in standard English?

Reviewer #1: No

Reviewer #2: Yes

5. Review Comments to the Author

Reviewer #1: Figures and some of the references were missing, so it was hard to review the manuscript. Moreover, this work presents some interesting results about a specific case study but it dose not have any theoretical contribution.

Reviewer #2: This paper based on mountain passenger flow data and built environment data,urban rail stations, this article divides stations into six categories, focuses on mountainous features such as the average longitudinal slope of roads, road network length, and road growth coefficient. It mainly examines walking accessibility around rail stations, transfer convenience, surrounding development intensity, and station passenger flow data. The K-means clustering method is applied to classify Chongqing’s rail transit stations. Subsequently, OLS, GWR, and MGWR regression models are established for different classifications to analyze the stations. Here are some of my suggestions.

In section 1

Question 1. Reference sorting and citation format.

Question 2. There is poor continuity between paragraphs, and the explanation of the three single-indicator classification approaches is vague.

Question 3. Pay attention to the paper alignment.

In section 2

Question 1. Pay attention to the spelling of words, e.g. H0 is "the", not "he".

Question 2. Pay attention to the format of the serial numbers of formulas 5, 6, and 7.

Question 3. Pay attention to the format of the subsequent tables on the spread.

Question 4. Where are the eleven indicators?Not thirteen?

Question 5. Pay attention to the part of speech of the table name, the "descriptive" in the second table should be changed to "description", and what exactly does the third column "abridge" refer to?

In section 3

Question 1. K represents the cluster type, so how can the K value be used to describe the number of clusters?

Question 2. Put the 189 stations after visualizing Chongqing in front of your analysis to observe easily.

Question 3. What are the parameters used to measure VIF and Moran'I,and how does the magnitude of their values affect the results?

Question 4. In the description of Table 4, what does the regression results of the two models of MGWR and GWR are all greater than 3,3 come from, and what does it mean, and what does it mean that the MGWR model can be based on GWR?

Question 5.What is the meaning of “**”in table 5?

Question 6. What are the references to independent variables, global indicators,and local indicators?

Question 7. Please describe Figure 4 as necessary.

Question 8.In section3.2.1 ,the last sentence is repeated, pay attention to the coherence of the sentence.

In section 4

Question 1. The content of the conclusion is cumbersome, and it briefly summarizes how to achieve this study, what conclusions have been reached, and how it can be applied to real life.

Question 2. If any, the shortcomings of the study and the future direction of the study should also be mentioned.

6. PLOS authors have the option to publish the peer review history of their article (what does this mean? ). If published, this will include your full peer review and any attached files.

**Do you want your identity to be public for this peer review?** For information about this choice, including consent withdrawal, please see our Privacy Policy .

Reviewer #1: No

Reviewer #2: No

---

## [Author Response · Author response to Decision Letter 1]

9 Jan 2025

Dear Editor and Reviewers,

Thanks very much for your time and valuable suggestions. The authors have carefully and seriously considered the questions and suggestions enclosed in the review and made necessary revisions.

Editor

(1)Response:

I'm sorry that the data was not complete before. The sorted data File has been uploaded to the system. The specific data is used as follows: The POI points near the original site extracted in this paper are in the support information S6 File, and the statistical data are in S2 File; The latitude and longitude of the site is in the support file S8. The attributes of the site are extracted in the following text, and the information around the site is based on the latitude and longitude of the site. Figure 1 use support information file S3, S4 code, based on the support files S1, S2, S6 data export. Figure 2 is based on support information S1 data. Figure 3 is based on supporting information S2 data. Figure 4 is derived based on the previously known classification and S8 location. Table 1 is based on support information S2 and S7. Table 2 shows descriptive expressions. Table 3 is based on support information S2,S6. Table 4 is based on supporting file information S2,S6,S1,S7.

Reviewer 1

(1)Response:We are sorry to bring you the confusion.As for the lack of references in the literature, we checked the full text again and found that some professional summaries were not cited. In this revision, 7 references have been added and cited, hoping to read the article better.

For the missing chart, we checked the full-text chart again, probably because Figure 4 was given directly without elaboration, leading to reading difficulties. Here we added an explanation of Figure 4.

The results of this paper can provide a basis for analyzing the passenger flow of the station in mountain cities, and provide a reference for the station to take measures to improve the passenger flow.

Modification: The new papers are added in section references. (Page22,24)

The text“Employment mountain sites are primarily distributed within the central urban areas, while employment non-mountain sites are located in the central regions of each group. Residential mountain sites are concentrated in the main urban area and its surroundings, whereas residential non-mountain sites are mainly distributed in the outskirts of each region, highlighting a pronounced separation between employment and housing. Comprehensive mountain sites are mainly situated along the South Bank and near Jiefangbei, while comprehensive non-mountain sites are predominantly located in Yuzhong and Shapingba districts. This distribution indicates that clusters closer to the urban center tend to have flatter terrain and a higher degree of land development.

Analyzing the development trajectory of administrative districts reveals potential factors contributing to the significant differences in the distribution of site types within the main urban area. The central urban group exhibits a higher degree of urbanization, concentrated development, and well-established urban functions, providing numerous employment opportunities and attracting employment-oriented sites. In contrast, development in groups outside the main city has progressed at a slower pace, with urban functions still under improvement and site development yet to fully take shape, resulting in the prevalence of residential sites in these areas. Furthermore, with regional centers being more developed than their peripheral counterparts, comprehensive stations tend to emerge in these clusters. Additionally, the unique geographical conditions and mountainous terrain of mountain cities significantly influence the spatial distribution of different station types.”is added to the paper.�Page 25

References 12,35,37,38,39,40,41 are cited in the article.(Page 5,13,16,17,18)

Reviewer 2

In section 1:

(1) Response: Thanks for your comments. Here, we reorganized the document ranking from small to large, and uniformly cited references at the end of the sentence.

Modification: The order of references and citations have been revised.Taking the last act of the first paragraph on page 3 as an example, we put reference 4 at the end of the sentence.(Page 3)

(2)Response: We are sorry to bring you the confusion. Here, we have reorganized the articulation at the beginning of each paragraph and restated the three individual indicators in more detail.For better connection�we divided the literature review into two parts: site classification and regression analysis. After site classification, the regression analysis of different sites can more intuitively show the impact of different factors on different types of sites, so that corresponding measures can be made for different types of sites to improve passenger flow.

Modification We have redivided site classification and regression analysis into two descriptions. At the begining of description, the “Since the characteristics of each station and its surrounding land use affect stations differently, leading to variations in passenger flow patterns, it is essential to investigate the mechanisms influencing passenger flow across different station types based on a reasonable classification of stations.” is added to the paper. The text “Wu Jiaorong (2007) categorized subway stations into distinct functional types based on their varying transfer facilities[9].” is added to the station characteristic indicators classfication. The text “In the United States, the Center for Transit-Oriented Development (2010) classified stations into 15 types by considering factors such as the number of jobs around the stations, the annual total mileage of households in the traffic area, and whether the surrounding areas were residential, employment-focused, or mixed-use [10].” is added to the land use classfication. The text “Meekyung (2018) grouped 233 subway stations in Seoul into eight categories using data on subway card swipes, passenger flow at different times, and fluctuations in passenger flow curves[11]. Similarly, Li et al. classified sites into six categories based on passenger traffic characteristics such as the number of peaks and troughs and the skewness in flow fluctuations [12].”is added to the passenger flow classfication. (Page4,5)

Two level 3 subheadings are added to the text. The subheading “1.1.1 Classification of subway stations”is added to the paper. (Page 4)

The subheading “1.1.2 Study on the influence mechanism of passenger flow”is added to the paper.(Page 5)

(3)Pay attention to the paper alignment.

Response: Thanks very much for your comment. To align the paper, this article has set the full page margins to 2cm to the left and right, and formatted the table to automatically resize according to the window.

In section 2:

(1)Response: We are sorry for the improper English expression. We have checked the full text and made extensive corrections for spelling, grammar and other errors. Only the part proposed by the reviewer is shown here.

Modification: The word “H0 is he”is revised as “H0 is the”. (Page 8)

(2)Modification: We adjusted the type of input method typing, and the “��” is changed to “()”(Page 12)

(3)Modification: Thanks for the reviewer’s comment. We have modified the table style of the full text to the format required by the journal, and the table size to adjust the size by window.

(4)Response: We are sorry for the mistakes. There should be 13 indicators here.

Modification: The "eleven" has been revised as "thirteen".(Page 9)

(5)Response: We are sorry for the problem. For this word error, we have made a correction in the article.The abridge stands for abbreviation

Modification: The "descriptive" has been revised as "description".The "abridge" has been revised as "abbreviation".�Page 11

In section 3

(1)Modification: The text “K-means clustering is an iterative analysis method that achieves optimal classification by continuously adjusting the positions of cluster centers [35]. The site data is divided into K groups, randomly select K objects as the initial cluster center, and then calculate the distance between each object and each seed cluster center, and assign each object to the nearest cluster center, the cluster center and the object assigned to them represent a cluster. For each sample assigned, the cluster center of the cluster is recalculated based on the existing objects in the cluster. This process is repeated until no objects are reassigned to different clusters.”is added to the paper. (Page 13)

(2)Response: Thanks for your comments.The analysis of 189 sites in Chongqing has been added to the back of the site visualization for easy review.(Page 14)

(3)Response: VIF is a parameter to test the correlation among influencing factors, Moran'I is a parameter to evaluate the significance of the influence, and the positive and negative values of Moran'I indicate that the influence is positively correlated or negatively correlated. The formulas for solving VIF and Moran'I parameter values are added in this paper.

Modification: “Only if there is no strict collinearity between the independent variables can the model analysis be conducted.The collinearity test is represented by the variance inflation factor (VIF).The greater the VIF, the higher the collinearity probability between the independent variable and other variables.VIF within 10 indicates that there is no serious collinearity[37].

The VIF of the jth independent variable is: formula(8)

Where:

VIFj is the variance expansion coefficient of the jth independent variable; Rj2 is the determination coefficient of the jth independent variable as the dependent variable, and the other independent variables are used as linear regression.

There may be spatial dependence between neighboring sites, and this influence cannot be explained by OLS regression, so it is necessary to test the spatial correlation between sites. In this study, Moran 'I was selected as the test index, and the spatial projection coordinates and influencing factors of each site were imported into Arc GIS for the spatial autocorrelation test[38].

The calculation formula of Moran'I is as follows:formula (9) (10)

Where:

m’ is the total number of rail transit stations;γij is the weight between sites;ei and ej are represented as independent variable indicators for the i and j sites;is the mean value of site independent variable indicator e;S2 is the variance of the site-independent variable index e;dij is the European distance between site i and site j;

b is the bandwidth, which refers to the non-negative decay parameter of a weighted remote function.

When the sample space distribution is relatively uniform, equal bandwidth is often used, and the opposite is used.The value of Moran’I is usually [-1, 1].Under certain significance tests, the Moran index is 0.That is, there is no spatial autocorrelation. A value greater than 0 indicates that the variable has agglomeration, while a value less than 0 indicates that the variable has dispersion (spatial negative correlation) [39]. Generally, normalized statistical Z-value is used to conduct a significance test, and its expression is as follows: formula (11)

Where:

E (Moran’I) is the theoretical mathematical expectation of the Moran’I; is the theoretical variance.

The reliability of Moran’I can be evaluated by the Z-value and P-value of significance. If the significance P value is less than 0.05 (through a 95% confidence test) the absolute value of the Z score exceeds the critical value of 1.96, indicating that the results of Moran’I are credible, and more than 95% of the certainty is that the data are spatially correlated. When the value of Moran’I is greater than 0, the spatial correlation is positive; when the value of Moran’I is less than 0, the correlation is negative[40].”The text is added to the article.(Page 15,16,17)

(4)Response: We are sorry for the unclear description. MGWR compared with the GWR model, the difference of indexes RSS, AIC, and AICc values is greater than 3, which means that MGWR model is better.The "3" here is derived from the paper "Research on the type identification and influence mechanism of urban rail transit stations from the perspective of passenger flow characteristics". For the sake of precise description, it is rephrased that MGWR significantly outperforms GWR in all indicators, thus making the model superior.Since both GWR and MGWR can be used to describe spatial variables, AICc is an important evaluation index to evaluate whether the model is true when describing spatial variables. The smaller AICc is, the more accurate the model regression is, and the AICc value of MGWR is less than GWR, so MGWR is considered to be the model optimized and improved based on GWR model.

Modification: The text “Taking the AICc index as the bandwidth optimization criterion, it is found that the AICc index value of the MGWR model regression is significantly smaller than that of the GWR model in different periods, which indicates that the MGWR model realizes further optimization based on the GWR model.” is added to the paper.(Page 19)

(5)Response: We are sorry for the unclear description. The "**" symbol indicates significance in the confidence interval.

Modification:The text“Note: ** indicates significance on a 95% confidence interval.”is added to the bottom of the table.(Page 20)

(6)Response: We apologize for the lack of references. We consulted the relevant data again, and quoted the data as reference in the paper.Just as references 41,42 are cited in the paper, these documents record the meaning of independent variables, global parameters, and local parameters in the model.

Modification: References 41,42 have been added on page 18.(Page 18)

(7)Response: We are sorry for the unclear description. We have added the description of Figure 4�seeing reviewer 1’s response for a detailed description of Figure 4.(Page 15)

(8)Response: We are sorry for the mistakes. There is a duplicate error in the statement, which has been modified in the article, and other statements have been checked for the same error.

Modification: The text has been revised as“The Z-values of the 12 indicators are above 1.96, achieving the level of significance,which meets the conditions for establishing a regression model.”

In section 4

(1)Response: Thanks very much for your comment. We took this advice and summarized the conclusions of this article more succinctly and applied them to real life, listing several improvements to increase traffic to the site.

Modification: The text has been revised as “Based on passenger flow data and built environment data from mountain urban rail stations, this study employs the K-means clustering algorithm to classify the stations and establishes OLS, GWR, and MGWR models for regression analysis of passenger flow characteristics. The findings indicate that stations can be categorized into six types: comprehensive non-mountain, comprehensive mountain, residential non-mountain, residential mountain, employment mountain, and employment non-mountain. A comparison of model performance reveals that the MGWR model achieves better regression fitting than the GWR and OLS models. The MGWR analysis highlights that the influence of various indicators on passenger flow is spatially non-stationary and varies with geographical location. For instance, a higher density of bus lines and parking lots around a site significantly increases peak passenger flow during the morning and evening. This effect is most pronounced at employment-oriented non-mountain stations and residential non-mountain stations. To enhance passenger flow, it is recommended to increase the number of buses and parking facilities near these types of stations. Additionally, lower average road slopes and road growth coefficients are associated with higher peak passenger flow, particularly at residential non-mountain stations. Thus, improving the pedestrian environment around these stations could attract more passengers.”. (Page 20,21

(2)Modification: The text “This study provides a theoretical foundation for increasing the modal share of rail transit, though certain limitations remain. The passenger flow data used in this study were collected at hourly intervals. To better understand

---

## [Decision Letter · Decision Letter 1]

25 Feb 2025

PONE-D-24-48526R1Classification of mountain-based rail transit stations and analysis of passenger flow influencing mechanismsPLOS ONE

Dear Dr. Zou,

Thank you for submitting your manuscript to PLOS ONE. After careful consideration, we feel that it has merit but does not fully meet PLOS ONE’s publication criteria as it currently stands. Therefore, we invite you to submit a revised version of the manuscript that addresses the points raised during the review process.

We look forward to receiving your revised manuscript.

Kind regards,

Qing-Chang Lu

Academic Editor

PLOS ONE

Reviewers' comments:

Reviewer's Responses to Questions

**Comments to the Author**

1. If the authors have adequately addressed your comments raised in a previous round of review and you feel that this manuscript is now acceptable for publication, you may indicate that here to bypass the “Comments to the Author” section, enter your conflict of interest statement in the “Confidential to Editor” section, and submit your "Accept" recommendation.

Reviewer #2: (No Response)

Reviewer #3: (No Response)

2. Is the manuscript technically sound, and do the data support the conclusions?

Reviewer #2: (No Response)

Reviewer #3: Partly

3. Has the statistical analysis been performed appropriately and rigorously? 

Reviewer #2: (No Response)

Reviewer #3: No

4. Have the authors made all data underlying the findings in their manuscript fully available?

Reviewer #2: (No Response)

Reviewer #3: Yes

5. Is the manuscript presented in an intelligible fashion and written in standard English?

Reviewer #2: (No Response)

Reviewer #3: Yes

6. Review Comments to the Author

Reviewer #2: (No Response)

Reviewer #3: I appreciate the opportunity to review this paper as an outside reviewer from the initial round of revisions. I send my gratitude to the authors, who worked hard to address the initial feedback that they received which helped me in conducting this review. They have made substantive improvements, most notably to the results section and should be commended for their efforts. This is an interesting topic (Mountain Transit) that has not been well considered in the research record.

However, upon my review of this manuscript, I found some worrying aspects of this research that the other reviewers did not previously discuss. I believe these need to be addressed. There are multiple areas in the data and methodology section where the authors did not justify their selection of method and parameters. This could negatively impact the reproducibility of this work.

I am deeply concerned about the use of OLS and GWR in this study on what I believe is discrete count data. I would like to provide the authors an opportunity to address this feedback, and to hopefully clear up some of my trepidations. For this reason, I have suggested major revisions at this time.

I offer the following comments, with a particular focus on the methodology section which needs the most clarification:

Introduction Section

Literature Review:

Page 5, section 1.1.1 — Please consider replacing the word ‘subway stations’ in your section header with ‘urban rail rapid transit stations’, as this better encapsulates above ground and below ground stations.

Study Area:

Page 8, section 1.2 — It is unclear to me whether the 10 referenced rail lines are all rapid transit lines, or if they include other rail modes such as commuter rail or light rail. This should be stated.

Data and Methodology Section:

Page 8, section 2.1 — As a reader, it would be helpful to understand why the authors chose the K-means clustering algorithm when there are other unsupervised classification methods available including DBSCAN, HDBSCAN, OPTICS, or Self-Organizing Maps (SOMs). Some studies have shown that these methods are more accurate classifiers than k-means, and the DBSCAN family in particular is used for spatial clustering which is often added as a component in transit studies. I do not think that the authors should have to use one of these methods, but it would be nice if they offered a deeper theoretical background justifying their choice of clustering method.

Page 9, section 2.1.1, 1st sentence — Why were these four indicators chosen? Please justify the selection criteria used in choosing these indicators.

Page 9, section 2.1.1, Indicator 1 — What is the data source for the POI data?

Page 9, section 2.1.1, Indicator 3 — Why was 500 meters chosen as the threshold value for the accessibility analysis? Is this based off of travel survey data or other measurements of passenger willingness to walk to a station?

Page 12, section 2.2.1, Indicator 1 — If this indicator is meant for “properties of the station itself”, then why would the indicator include “whether [a station] is close to a mega commercial area” as a factor? This is a built environment variable. Also, what constitutes a “large transfer station”? Is this based off of passenger flow statistics?

Page 13, section 2.2.2 — As a reader I would personally find it more helpful for the authors to state the independent and dependent regression variables being used in the modeling. It is not all that helpful to show a generic OLS formula. I would like to see how this formula is being applied by the authors to advance the research question. I provide the same feedback for the GWR model, all variables should be discussed in the context of the additional spatial component of the model.

Page 13, section 2.2.2 — Passenger flow data, to my understanding, is a form of discrete count data demonstrating the number of passengers boarding or alighting at a station. Such data should generally be modeled using count models, such as those in the Poisson family (Poisson, Negative Binomial) or their spatial counterparts. Can the authors please justify their selection of OLS and its spatial counterparts (GWR and MGWR)? The authors only cite one example of OLS being used by Loo et al. in the literature review. The properties of the flow data is also not deeply explained in section 2.2.1. Perhaps it is not a count type data, which would then justify the use of OLS. I would appreciate clarity from the authors on this subject.

Analysis of Results Section

Page 15, section 3.1, cluster 1 — What are ‘composite-type’ stations? This is a very unusual terminology. From my understanding, these stations have the highest passenger volumes, but I am not sure what the word ‘composite’ has to do with passenger volumes.

Pages 17-20, section 3.2.1 — The authors should have explained their use of VIF and Moran’s I as diagnostics in the methods section, it is unusual that these formulas are being introduced and discussed methodologically in the results section. Also, the use of European Distance Band for the Moran’s I needs to be justified as this can have major implications on the precision of the findings.

The corrections and adjustments to the results and conclusion are very appreciated.

7. PLOS authors have the option to publish the peer review history of their article (what does this mean? ). If published, this will include your full peer review and any attached files.

**Do you want your identity to be public for this peer review?** For information about this choice, including consent withdrawal, please see our Privacy Policy .

Reviewer #2: No

Reviewer #3: No

---

## [Author Response · Author response to Decision Letter 2]

28 Feb 2025

On behalf of all the contributing authors, I would like to express our sincere appreciation of your letter and reviewers' constructive comments concerning our article entitled Classification of mountain-based rail transit stations and analysis of passenger flow influencing mechanisms (Manuscript ID PONE-D-24-48526)I have considered the comments very carefully and have revised the paper accordingly.

These comments are all valuable and helpful for improving our article. We have extensively modified our manuscript according to the editor and reviewers' comments.We hope that the corrections are satisfactory.

---

## [Decision Letter · Decision Letter 2]

17 Apr 2025

Classification of mountain-based rail transit stations and analysis of passenger flow influencing mechanisms

PONE-D-24-48526R2

Dear Dr. Zou,

We’re pleased to inform you that your manuscript has been judged scientifically suitable for publication and will be formally accepted for publication once it meets all outstanding technical requirements.

Kind regards,

Qing-Chang Lu

Academic Editor

PLOS ONE

Additional Editor Comments (optional):

Reviewers' comments:

Reviewer's Responses to Questions

**Comments to the Author**

1. If the authors have adequately addressed your comments raised in a previous round of review and you feel that this manuscript is now acceptable for publication, you may indicate that here to bypass the “Comments to the Author” section, enter your conflict of interest statement in the “Confidential to Editor” section, and submit your "Accept" recommendation.

Reviewer #3: All comments have been addressed

2. Is the manuscript technically sound, and do the data support the conclusions?

Reviewer #3: Yes

3. Has the statistical analysis been performed appropriately and rigorously? 

Reviewer #3: Yes

4. Have the authors made all data underlying the findings in their manuscript fully available?

Reviewer #3: Yes

5. Is the manuscript presented in an intelligible fashion and written in standard English?

Reviewer #3: Yes

6. Review Comments to the Author

Reviewer #3: I would like to commend the authors for addressing the revisions that were suggested to them in the first round and also for deeply considering my own comments in the second round of revisions.

Having read the edits and the authors' responses to my comments, I feel confident that they have added the missing level of methodological detail that I described to them previously. Their explanation for choosing OLS and K-means is also appreciated, and the reorganized manuscript makes intuitive sense. In combination, these edits should make the study more impactful and replicable for scholars interested in urban rail.

At this time, considering the efforts of the authors and increased quality of the manuscript, I would recommend this work for publication in PLOS One.

7. PLOS authors have the option to publish the peer review history of their article (what does this mean? ). If published, this will include your full peer review and any attached files.

**Do you want your identity to be public for this peer review?** For information about this choice, including consent withdrawal, please see our Privacy Policy .

Reviewer #3: No

---

## [Editor Report · Acceptance letter]

PONE-D-24-48526R2

PLOS ONE

Dear Dr. Zou,

I'm pleased to inform you that your manuscript has been deemed suitable for publication in PLOS ONE. Congratulations! Your manuscript is now being handed over to our production team.

Kind regards,

on behalf of

Dr. Qing-Chang Lu

Academic Editor

PLOS ONE